# The virulence regulator CovR boosts CRISPR-Cas9 immunity in Group B *Streptococcus*

Adeline Pastuszka [1,2,6], Maria-Vittoria Mazzuoli [3,6], Chiara Crestani [4], Léonie Deborde [3], Odile Sismeiro[3], Coralie Lemaire[1,2], Vanessa Rong[1], Myriam Gominet[3], Elise Jacquemet [5], Rachel Legendre [5], Philippe Lanotte [1,2,7] ✉ & Arnaud Firon [3,7] ✉

CRISPR-Cas9 immune systems protect bacteria from foreign DNA. However, immune efficiency is constrained by Cas9 off-target cleavages and toxicity. How bacteria regulate Cas9 to maximize protection while preventing auto-immunity is not understood. Here, we show that the master regulator of virulence, CovR, regulates CRISPR-Cas9 immunity against mobile genetic elements in *Streptococcus agalactiae*, a pathobiont responsible for invasive neonatal infections. We show that CovR binds to and represses a distal promoter of the *cas* operon, integrating immunity within the virulence regulatory network. The CovR-regulated promoter provides a controlled increase in off-target cleavages to counteract mutations in the target DNA, restores the potency of old immune memory, and stimulates the acquisition of new memory in response to recent infections. Regulation of Cas9 by CovR is conserved at the species level, with lineage specificities suggesting different adaptive trajectories. Altogether, we describe the coordinated regulation of immunity and virulence that enhances the bacterial immune repertoire during host-pathogen interaction.

CRISPR-Cas9 is an adaptive immune system protecting bacteria against invading DNA, such as bacteriophages and other mobile genetic elements (MGEs)[1]. The Cas9 nuclease together with two non-coding RNAs, the tracrRNA and the crRNA, ensures sequence-specific recognition and cleavage of foreign DNA[2]. Given its nuclease activity, Cas9 must be tightly controlled to prevent collateral DNA cleavage. To date, CRISPR-Cas9 is assumed to be an autonomous system transcribed independently of physiology, stress, or any specific bacterial factor. One safeguard mechanism is an intrinsic feedback loop mediated by a long form of the tracrRNA that guides Cas9 to repress its own transcription[3]. Loss of *cas9* autorepression increases Cas9 expression and enhances immunity but reduces fitness in the absence of bacteriophages[3–6],

highlighting the trade-off between protection and Cas9 toxicity due to off-target cleavage.

The need to constrain auto-immunity and toxicity is shared among the distinct types of prokaryotic defence systems, in all the diversity of their composition and mechanisms. Viruses and other selfish MGEs exploit this vulnerability to inhibit immunity primarily at the post-translational level[7]. In turn, bacteria have diversified their immune systems and have evolved inducible defences. One straightforward mechanism involves transcriptional regulation, as observed in type I and III CRISPR systems in Gram-negative bacteria[7,8]. Transcriptional regulation, triggered by quorum sensing or membrane stress, provides need-based protection in high-risk environments while minimizing associated costs[9–12].

[1]Université de Tours, INRAE, UMR, 1282 ISP Tours, France. [2]CHRU de Tours, Service de Bactériologie-Virologie, Tours, France. [3]Institut Pasteur, Université Paris Cité, Department of Microbiology, Paris, France. [4]Institut Pasteur, Université Paris Cité, Biodiversity and Epidemiology of Bacterial Pathogens, Department of Global Health, Paris, France. [5]Institut Pasteur, Université Paris Cité, Bioinformatics and Biostatistics Hub, Paris, France. [6]These authors contributed equally: Adeline Pastuszka, Maria-Vittoria Mazzuoli. [7]These authors jointly supervised this work. Philippe Lanotte, Arnaud Firon. ✉e-mail: philippe.lanotte@univ-tours.fr; arnaud.firon@pasteur.fr

In this study, we characterized the type II-A CRISPR-Cas9 system of *Streptococcus agalactiae* (GBS: Group B *Streptococcus*), a pathobiont responsible for most invasive infections in the first three months of life[13,14]. The CRISPR-Cas9 locus is almost identical to the canonical system of *Streptococcus pyogenes* (GAS: Group A *Streptococcus*)[15,16]. The feedback loop mediated by the long tracrRNA, which limits Cas9 expression, is also conserved in both species[3]. However, it was noticed that some GBS strains have lost the constitutive and autorepressed *cas9* promoter, hereafter referred to as P1$_{cas}$, following chromosomal deletion[3]. Despite P1$_{cas}$ deletion, CRISPR-Cas9 immunity is functional due to a distal promoter, hereafter referred to as P2$_{cas}$, located at the 3' end of the tracrRNA[17]. In the P1$_{cas}$ deleted strains, the activity of the P2$_{cas}$ promoter (formerly the *srn036* promoter) is necessary and sufficient for *cas* transcription and to confer an effective immune response[17]. Remarkably, strains having lost the P1$_{cas}$ promoter belong to the CC-17 hypervirulent lineage responsible for most of neonatal meningitis[18–20].

Here, we show that the P2$_{cas}$ promoter is directly regulated by the master regulator of virulence CovR. The two-component system CovRS directly represses an array of virulence factors and is involved in almost every stage of host-pathogen interactions, from colonization to systemic infections[21–24]. Inactivation of CovR, *i.e* the release of CovR

repression, increases pathogenicity by unleashing a storm of toxins and adhesins[21–24]. The co-optation of the immune response in the virulence regulatory network through the repression of the P2$_{cas}$ promoter provides an inducible mechanism to strengthen immunity. The boost in immunity protects cells against invaders with mutated sequences that would otherwise escape recognition and stimulates both the recall of old memory against past infections and the acquisition of new memory against current infections.

## Results

### CovR directly represses cas9 transcription

In GBS, the *cas* operon is transcribed from the autorepressed P1$_{cas}$ promoter and from the distal P2$_{cas}$ promoter (Fig. 1A). The P2$_{cas}$ promoter is conserved while the P1$_{cas}$ promoter is deleted in the hypervirulent CC-17 lineage (Fig. 1A)[17]. Published genome-wide binding data in two wild-type strains[21], BM110 (CC-17) and NEM316 (CC-23), showed CovR binding 300 bp upstream of *cas9*, corresponding to the conserved P2$_{cas}$ promoter region (Fig. 1B). The ChIP-seq signal at the *cas* locus is among the most significant in both strains and is comparable to the binding signals at the promoters of virulence genes directly repressed by CovR[21]. Using purified recombinant protein, we confirmed in vitro the binding of CovR to the P2$_{cas}$ promoter (Fig. 1C).

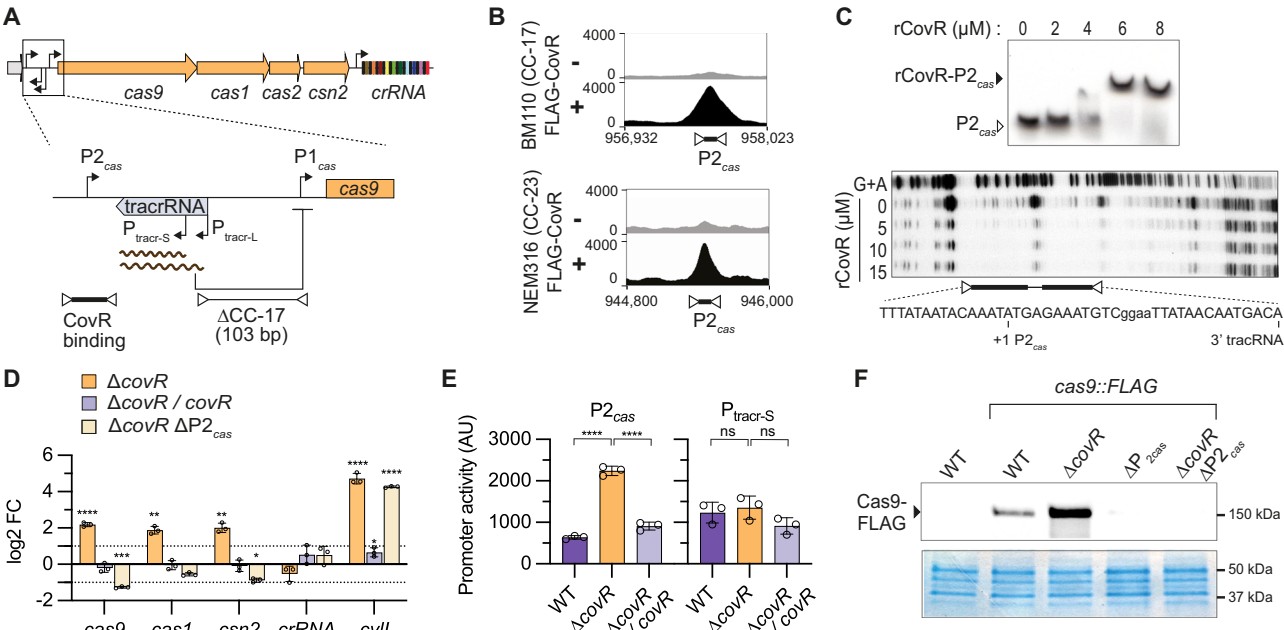

**Fig. 1 | CovR represses *cas* transcription via the P2$_{cas}$ distal promoter.**
**A** Organization of the CRISPR-Cas9 chromosomal locus. Upper: the *cas* operon and the crRNA encoded by the CRISPR array are organized in two transcriptional units. Lower: zoom-in on the upstream region of *cas9* containing the tracrRNA with its short and long promoters (P$_{tracr-S}$ and P$_{tracr-L}$) and the two *cas* promoters (P1$_{cas}$ and P2$_{cas}$). The tracr-L long transcript represses *cas* transcription by binding to the P1$_{cas}$ promoter. The absence of the P1$_{cas}$ promoter following a chromosomal deletion in CC-17 strains is highlighted, as well the region corresponding to CovR binding identified by ChIP-seq. B CovR binds to the P2$_{cas}$ promoter in vivo. ChIP-seq profiles of FLAG-tagged CovR at the P2$_{cas}$ loci in two WT strains (BM110 and NEM316) in non-inducing (-) and inducing condition (+). **C** CovR binds to the P2$_{cas}$ promoter in vitro. Upper panel: Electrophoretic mobility shift assay with recombinant rCovR showing delayed migration of the rCovR-P2$_{cas}$ complex compared to the free P2$_{cas}$ radiolabelled probe. Lower panel: DNase I protection assay with increasing rCovR concentrations. The in vitro rCovR-protected sequence is highlighted, along with the P2$_{cas}$ transcriptional start site (+1) and the 3' end of the tracrRNA. Source data are provided as a Source Data file. (D) CovR represses *cas* transcription. Gene expression by RT-qPCR relative to the BM110 strain in the Δ*covR* mutant (orange), the Δ*covR/covR* complemented strain (light violet), and the double Δ*covR* ΔP2$_{cas}$

mutant (light yellow) in which the −10 box and +1 TSS of P2$_{cas}$ were mutated. In addition to CRISPR-*cas* genes (*cas9, cas1, csn2,* and *crRNA*), expression of the *cyII* gene directly repressed by CovR and required for toxin synthesis is provided as control. Bars represent the mean ± SD of biological replicates (*n* = 3). Significant P-value from multiple two-tailed unpaired t-tests (FDR 1%, two-stage method of Benjamini- Krieger and Yekutieli) against the WT strain are highlighted with stars (****P < 0.0001; *** <0.001; ** <0.01; * <0.05). Source data, including exact P-value, are provided as a Source Data file. (E) CovR regulates P2$_{cas}$ activity. The P2$_{cas}$ and P$_{tracr}$ promoters were cloned upstream of a ß-galactosidase reporter in the pTCV-lac vector. Promoter activity was quantified in BM110 (WT: dark violet) and mutants using colorimetric assays. Bars represent the mean ± SD of biological replicates (*n* = 3). Significance was determined by unpaired, parametric, two-tailed t-test (****P < 0.0001; ns not significant). Source data, including exact P-value, are provided as a Source Data file. **F** CovR represses Cas9 expression. The FLAG epitope coding sequence was introduced in-frame at the 5' end of *cas9* (*cas9::FLAG*) in BM110 (WT) and in Δ*covR*, ΔP2$_{cas}$, and Δ*covR* ΔP$_{cas}$ mutants. Upper: representative Western blot from a biological duplicate (*n* = 2) using total protein extract and anti-FLAG antibody. Lower: corresponding Coomassie staining of total proteins used as loading control. Source data are provided as a Source Data file.

Electrophoretic mobility shift assays showed delayed migration of the 243 bp radiolabelled P2$_{cas}$ probe upon complexing with CovR, and DNase I protection assay narrowed down the CovR binding site to a sequence at the 3′ end of the tracrRNA encompassing the P2$_{cas}$ transcriptional start site (Fig. 1C).

To test if CovR regulates *cas* transcription, we first used the BM110 strain in which CRISPR-Cas9 immunity is solely dependent on P2$_{cas}$[17]. Quantitative PCRs after reverse transcription (RT-qPCR) showed a 4-fold increase of *cas9, cas1* and *csn2* transcription in the Δ*covR* mutant compared to the WT strain (Fig. 1D). Chromosomal complementation, obtained by reintroducing *covR* at its locus in the Δ*covR* mutant, restored *cas* transcription to WT levels (Fig. 1D). In addition, *cas* overexpression was not observed in a Δ*covR* ΔP2$_{cas}$ double mutant in which the P2$_{cas}$ is inactivated by targeted mutagenesis (7 SNPs substituting the −10 box and +1 transcriptional start site[17]), confirming that CovR acts through P2$_{cas}$ (Fig. 1D). As controls, we quantified transcription of the crRNA, which is transcribed independently of the *cas* operon, and of the CovR-regulated *cylI* gene necessary for the synthesis of the ß-h/c toxin[21,25]. As expected, crRNA transcription was CovR and P2$_{cas}$ independent, while transcription of *cylI* increased in the Δ*covR* mutant (Fig. 1D).

To test promoter activities, we employed a β-galactosidase transcriptional reporter system. Increased promoter activity in the Δ*covR* mutant was observed with the P2$_{cas}$ but not P$_{tracr}$ (Fig. 1E), confirming CovR-dependent regulation of the P2$_{cas}$ promoter only. Next, we evaluated Cas9 expression by introducing a FLAG epitope-tag sequence at the 5′ end of *cas9* into the chromosome. Analysis of total protein extracts with anti-FLAG antibodies confirmed Cas9 overexpression in the Δ*covR* mutant (Fig. 1F). Altogether, these results demonstrate that CovR binds to the P2$_{cas}$ promoter and represses transcription of the *cas* operon.

## CovR inactivation increases CRISPR-Cas9 immunity

Cas9 recognizes and cleaves foreign DNA with sequences (*i.e.*, protospacers) identical to those of the spacers present in the CRISPR array. To test immunity, we introduced protospacers into the replicative pTCV vector and quantified the number of kanamycin-resistant bacteria after GBS transformation (Fig. 2A). We then calculated an immunity index ($R_T$) as the ratio of transformation efficiency (transformants/µg vector) obtained with vectors containing a protospacer to that obtained with an empty vector. An immunity index of 1 (*i.e.*, Log2 $R_T = 0$) reflects no immunity, while an $R_T < 1$ (*i.e.*, Log2 $R_T < 0$) indicates recognition and cleavage of the protospacer-containing vectors (Fig. 2A).

The BM110 strain has 13 unique spacers in its CRISPR array. We first tested CRISPR-Cas9 immunity using 5 corresponding protospacers. As expected, no transformants were obtained in the WT strain with vectors containing protospacers corresponding to spacers at position 1, 4, and 8 in the array (Fig. 2B). Contrarily, vectors containing protospacers 12 and 13 led to immune deficiency with a $-3 <$ Log2 $R_T < -1$, corresponding to a two- to ten-fold reduction in the number of transformants compared to the empty vector (Fig. 2B). Remarkably, we observed full immunity (Log2 $R_T < -10$) against the 5 protospacers in the Δ*covR* mutant, including protospacers 12 and 13 (Fig. 2B). Mutations of the protospacer adjacent motif (PAM), which is essential for discriminating between self and non-self DNA[1,26], abolished immune recognition and cleavage in both the WT and Δ*covR* mutant across all tested protospacers (Fig. 2B). Considering that spacers are acquired sequentially at the leader end of the CRISPR array[27,28], these results reveal that inactivation of *covR* enhances immune protection against older spacers, thereby protecting from vanishing immune memory.

Next, we introduced single mismatches in the protospacer 8 at the PAM-proximal positions 1 to 14 (Fig. 2B). As shown in Fig. 2C, a reciprocal transversion (A ↔ T or G ↔ C) at position 1 to 11 abolished or diminished ($-4 <$ Log2 $R_T ≤ 0$) immune efficiency in the WT strain

(Fig. 2C). However, inactivation of *covR* maintained near full immunity against mutated protospacers at position 3, 5, 6, 9, 10, and 11 (Fig. 2C). Similar immunity levels were observed between the WT and Δ*covR* mutant for other positions that tolerated (position 12, 13, and 14) or not (position 1, 2, 4, 7, and 8) mutations (Fig. 2C).

Next, we tested consecutive double mismatches in protospacer 4. Double mutations close to the PAM motif are not tolerated in both WT and Δ*covR* mutant and result in a loss of immunity (Fig. 2D). Internal double mutations have a variable effect in the WT strain depending on their position, reducing, or abolishing immunity except for position 15–16 (Fig. 2D). Strikingly, inactivation of *covR* maintains almost full immune recognition and cleavage of internal protospacers containing double mismatches (Fig. 2D). Complementation of the Δ*covR* mutant restored a WT level of immunity against older and mutated protospacers, and inactivation of the P2$_{cas}$ promoter abolished immunity in the Δ*covR* mutant (Fig. 2E). These results show that CovR regulation of the P2$_{cas}$ promoter protects against the loss of immune memory and enhances protection against mutated protospacers.

## Gain in immunity is concentrated in the seed region

To systematically test the effect of mismatch, we introduced all possible single mutations (A/T/G/C) at each position in protospacers 4 and 8, creating a pool of mutated vectors (Input pool) which were introduced into GBS (Fig. 3A). After selection, kanamycin-resistant transformants were pooled, vectors were purified (Output pools), and inserts were amplified by PCR and sequenced by Illumina (Fig. 3A). We confirmed that the input pool contains an over-representation of the P4 and P8 protospacers (one-quarter of possible mutation restoring WT protospacer sequence), a similar proportion of each of the 120 mutated protospacers, and 3 protospacers with random sequences that were added in low amounts (Supplementary Data S1). After GBS transformation, the WT protospacer sequences were depleted (positive controls) and protospacers with random sequence were enriched (negative controls) in the WT and Δ*covR* mutant, but not in the Δ*covR* ΔP2$_{cas}$ double mutant, which does not express Cas9, validating the screening condition for bulk immune selection (Supplementary Fig. S1A).

Single mutations in protospacers P4 and P8 impact immune efficiency and can be divided into two main groups depending on their position. Mutations from positions 20 to 9 are strongly depleted in both the WT and Δ*covR* mutant, with exceptions, while mutations from positions 8 to 1 strongly abolished immunity in the WT but showed variable effects in the Δ*covR* mutant (Fig. 3B). To highlight the advantage conferred by *covR* inactivation, we quantified the fold change for each single mutation relative to the WT strain. Significant immune gains extend to the seed region, from base one to eight, the region of the gRNA that initiates base pairing and R-loop formation (Fig. 3C). Nevertheless, effects of protospacer mutations on immunity depend on spacer sequence, base position, and type of mutation. The highest difference between the WT and the Δ*covR* mutant was observed for mutations between positions 4 and 6 (Fig. 3C and Supplementary Fig. S1), corresponding to the sequence immediately downstream of the Cas9 cleavage sites.

In contrast to most two-component systems, CovR functions as a repressor, and the expression of its regulated genes depends on its inactivation. To investigate the role of the cognate histidine kinase CovS in immune efficiency, we repeated the bulk immunity assay using a phosphoablative CovR mutant (CovR D$_{53}$A) and a hyperphosphorylated CovR mutant (CovS T$_{282}$A, which specifically abolishes the phosphatase activity of CovS[21,29,30]). The inactive CovR D53A variant, which cannot be phosphorylated by CovS, exhibited increased immunity against single mismatches in the seed region of protospacers 4 and 8 (Supplementary Fig. S2). This profile resembles that of the Δ*covR* mutant, albeit with attenuated immune depletion. This suggests that the Stk1 serine/threonine kinase, which antagonistically

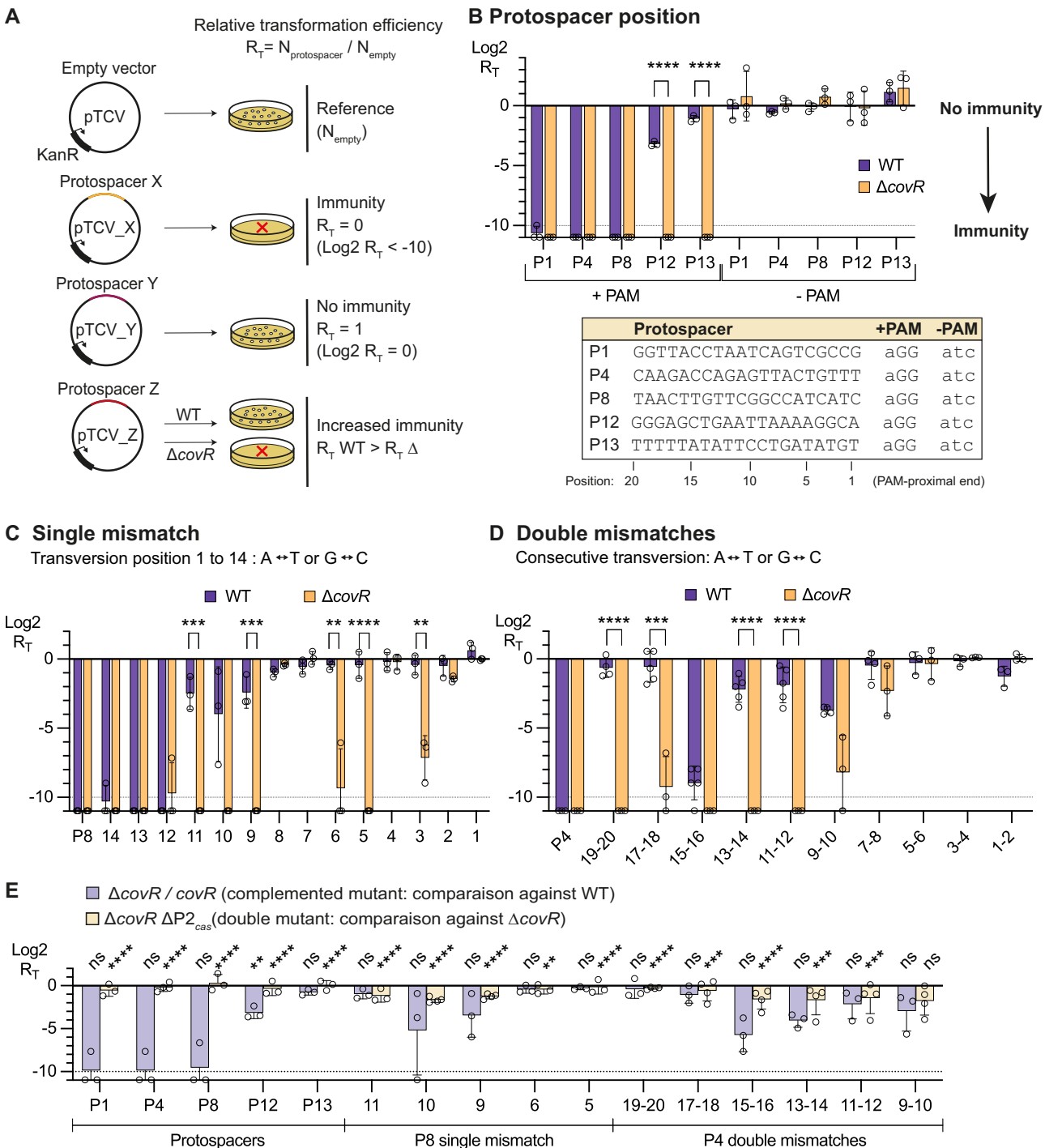

**Fig. 2 | CovR inactivation increases the immune repertoire. A** CRISPR immunity assay. Protospacers (X, Y, or Z) corresponding to spacer sequences in the genomic CRISPR array are cloned into the shuttle vector pTCV and transformed into GBS strains. The number of kanamycin resistant transformants by μg of empty vector ($N_{empty}$) is used as reference to infer the relative transformation efficiency for each protospacer ($R_T$). Immunity results in cleavage of the vector and the absence of transformants ($R_T = 0$), while ineffective binding or cleavage of the protospacer results in a number of transformants similar to that obtained with the empty vector ($R_T = 1$). The increase in immune repertoire refers to a difference in $R_T$ between the WT and the Δ$covR$ mutant for a given protospacer. **B** Immunity depends on spacer position and CovR inactivation. Immunity assays (Log2 $R_T$) in BM110 strain (WT, dark violet) and Δ$covR$ mutant (orange) with protospacers corresponding to the first (P1), internal, (P4, P8, P12), and last (P13) spacers in the CRISPR array. Controls without a protospacer adjacent motif (-PAM) are included. Full immunity is defined

as no transformants (below the limit of detection of $10^3$ transformants per μg) and is represented by a dashed line (Log2 = −10). **C** CovR inactivation enhances immunity against single-point mutation. Similar immunity assays using single mutations (transversion) introduced into the P8 protospacer sequence at the PAM-proximal position 1 to 14. **D** CovR inactivation enhances immunity against mutated protospacers. Similar immunity assays using consecutive mutations in the P4 protospacer. **E** CovR acts through the P2$_{cas}$ promoter. Immunity assays with non-optimal spacers in the Δ$covR$/$covR$ complemented strain (light violet) and the Δ$covR$ ΔP$_{cas}$ double mutant (light yellow). For individual immunity assays (**B**–**E**), bars represent the mean and error bars the SD of at least three biological triplicates ($n = 3$). Significant $P$-values from multiple two-tailed unpaired $t$-tests (FDR 1%, two-stage method of Benjamini, Krieger, and Yekutieli) are highlighted with stars (****$P < 0.0001$; ***$P < 0.001$; **$P < 0.01$; *$P < 0.05$; ns, not significant). Source data, including exact $P$-value, are provided as a Source Data file.

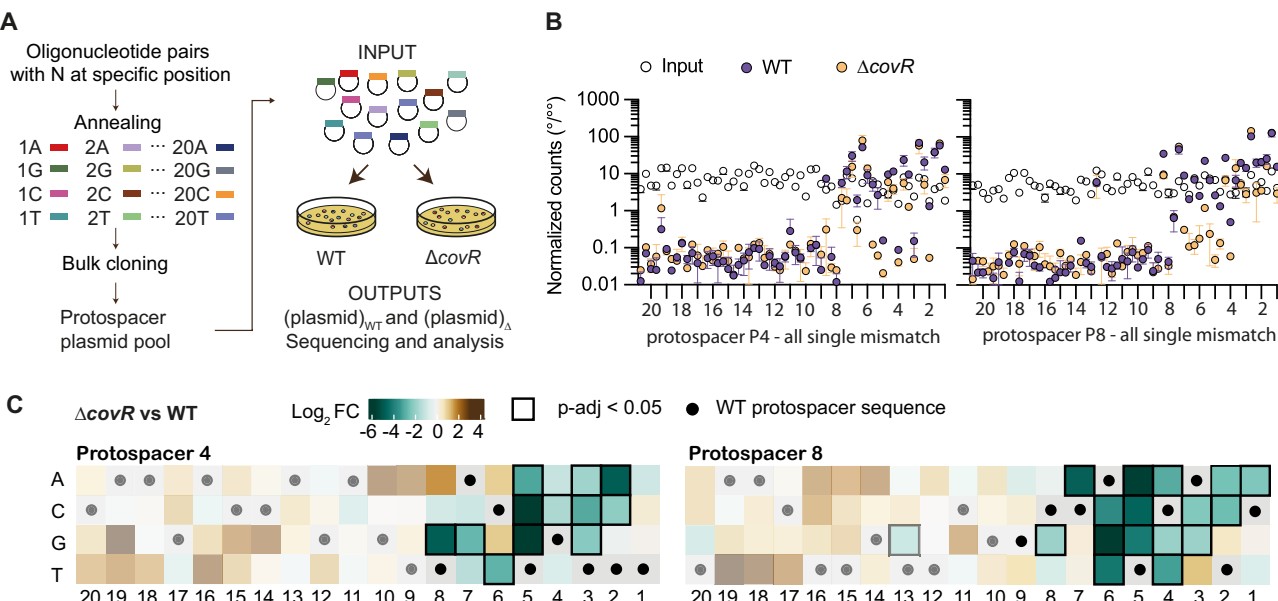

**Fig. 3 | CovR inactivation confers tolerance to mutations in the seed region.**
**A** Bulk CRISPR immunity assay. Synthetic protospacers were designed and cloned en masse to obtain a pool of plasmids with all possible single mutations at each position. The plasmid pool (INPUT) was transformed into the wild-type strain BM110 (WT) and the ΔcovR mutant. Kanamycin-resistant GBS transformants were then pooled and plasmid extracted (OUTPUT pools). Vector inserts were amplified by PCR, and the amplicons sequenced by Illumina to quantify the proportion of each protospacer sequence in each condition. **B** Bulk CRISPR immunity assay. Normalized counts of all possible single mutations at each position of the P4 and P8 protospacer in the input pool (white dots) and after transformation in the WT (blue dots) or ΔcovR mutant (orange dots). Each dot represents the mean +/− SD of biological replicate for the WT and ΔcovR mutant ($n = 2$), and of technical replicate ($n = 3$ plasmid purification) for the input pool. **C** CovR inactivation increases immunity to mutated seed regions. Gain in immunity ($\log_2$ fold change: green to brown) for the ΔcovR mutant compared to the WT strain for all possible single mutations in the P4 and P8 protospacers. The wild-type spacer sequences are depicted with a dot at each position. Statistical analysis with DESeq2 uses the Wald test to compute p-values, followed by Benjamini-Hochberg correction to adjust for multiple comparisons. Significances (p-adj <0.05) are highlighted with black square borders. Raw count, p-values, and p-adj are provided in Supplementary Data S1.

phosphorylates CovR at threonine 65[31] is required for full CovR inactivation (Supplementary Fig. S2). Conversely, hyperactivation of CovR due to the loss of CovS phosphatase activity did not significantly affect immune efficiency (Supplementary Fig. S2). This is consistent with previous analyses showing that CovR is already phosphorylated under the tested conditions (i.e., THY-rich medium) and that CovR hyper-phosphorylation only marginally affects gene repression due to its already near-maximal repression at steady state[21,30]. Overall, CovR inactivation primarily enhances Cas9 activity against mutations in the PAM-proximal seed region, the key determinant of Cas9 binding specificity. The P2$_{cas}$ promoter is repressed by CovR, maintaining a minimal level of *cas* transcription necessary for immune efficiency. CovR inactivation relieves this repression, expanding the immune repertoire against related sequences through a mechanism involving CovS and likely Stk1.

## CovR promotes the acquisition of new memory

Cas9 mediates an adaptive immune response relying on the acquisition of immune memory from past infections. We tested the acquisition of new spacers by an established plasmid challenge assay using the replicative pNZ123 vector (Fig. 4A)[1,17]. Acquisition of a new spacer occurs at the leader end[27,28] and led to a 66 bp-increase of the CRISPR locus size (30 bp spacer + 36 bp repeat) within the array that can be detected in the bacterial population by PCR (Fig. 4B). After GBS transformation with pNZ123 and chloramphenicol selection, isolated transformants were inoculated in a rich medium without antibiotic selective pressure, grown until saturation, and subcultured for nine consecutive passages. From the first passage, the proportion of bacteria with N + 1 spacers in the ΔcovR population was higher than in the WT and complemented strains (Fig. 4C). By the third passage, most cells in the ΔcovR population already have N + 1 spacers, while the proportion of cells with N + 1 spacers in the WT and complemented

strains reaches roughly one quarter only by the end of the experiment (Fig. 4C). Therefore, the increased transcription of the *cas* operon in the absence of CovR promotes adaptive immunity through the acquisition of new memory.

## CovR-regulated immunity is conserved in the GBS population

Thus far, we have demonstrated the regulation of CRISPR-Cas9 immunity by CovR in a CC-17 strain that does not have the constitutive P1$_{cas}$ promoter. The absence of the P1$_{cas}$ promoter in some strains was previously noticed but not analysed at the population level[3]. Using 1069 genomes representative of the diversity of the GBS population, grouped into sequence lineage (SL) and clonal group (CG) by genome-based phylogeny (gMLST) analysis[32], we confirmed the absence of the P1$_{cas}$ promoter in the hypervirulent human lineage CG-17 (Fig. 5A). The P1$_{cas}$ promoter is also absent in the distant CG-103, −314, −612, and −609 clades, which encompass generalist lineages mainly responsible for dairy cattle and camel infections, but very rarely in other lineages (Fig. 5A). Loss of the P1$_{cas}$ promoter is associated with specific Cas9 variants (Fig. 5B), suggesting an adaptation that could either increase or decrease Cas9 processivity and fidelity in the absence of P1$_{cas}$. In addition to P1$_{cas}$ presence/absence and Cas9 variants, a third variable is the steady-state level of CovR activation, which varies between strains[21] and thus could impact the basal level of *cas* repression.

To test the conserved regulation of Cas9 by CovR, we deleted *covR* in five additional representative isolates. RT-qPCR confirmed the four-fold activation of *cas9* and *csn2* in a second SL-17 strain, COH1 (Fig. 5C). However, only a modest increase (~2-fold) was observed in the NEM316 (SL-23), 515 (SL-23), 2603V/R (SL-110), and A909 (SL-7) backgrounds (Fig. 5C). Similar overexpression of the CovR-regulated *cylE* gene was observed in ΔcovR mutants, except for an attenuated response in A909 (Fig. 5C), suggesting that the presence of the P1$_{cas}$

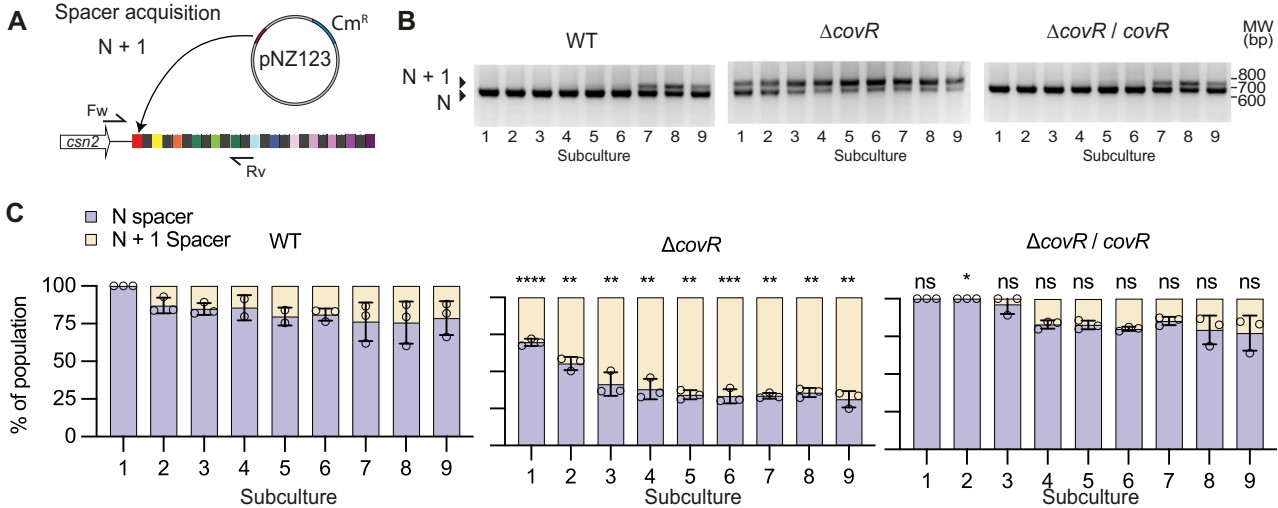

**Fig. 4 | CovR inactivation promotes adaptive immunity. A** Spacer acquisition assay. Acquisition of a new spacer at the leader end of the CRISPR array is tested by PCR with specific forward (Fw) and reverse (Rv) primers. Spacer acquisition was monitored in the population over 9 serial subcultures without selective antibiotic pressure to maintain the pZN123 vector. **B** Representative PCRs showing products corresponding to N and N + 1 spacer (+66 bp) over time in the WT, Δ*covR* mutant, and Δ*covR*/*covR* complemented strain. Source data are provided as a Source Data file. **C** Quantification of new spacer acquisition rate. Proportion of N (light violet) and N + 1 (light yellow) spacers in the bacterial population are quantified by densitometry of PCR products. Data represent the mean ± SD of biological replicates (*n* = 3). Significance against the WT for each subculture was determined by unpaired, parametric, two-tailed t-tests (****$P < 0.0001$; ***$P < 0.001$; **$P < 0.01$; *$P < 0.05$; ns, not significant). Source data, including PCR gels and exact *P*-value, are provided as a Source Data file.

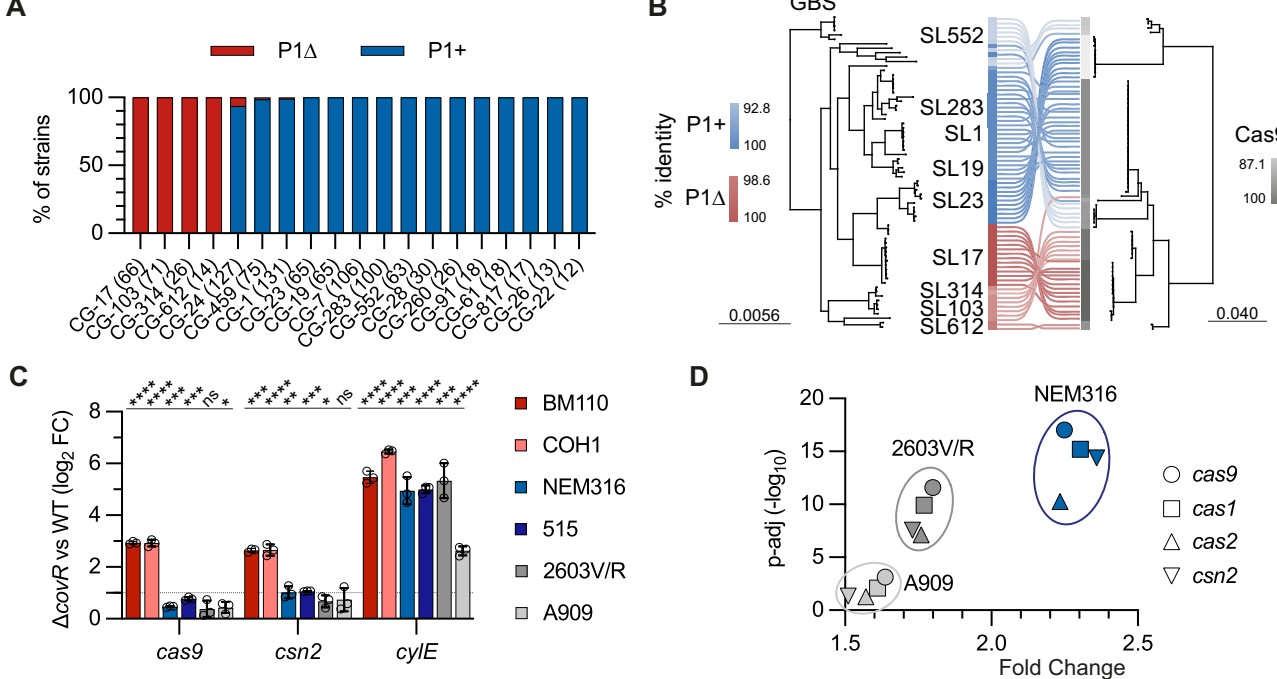

**Fig. 5 | The P1_cas promoter dampens CovR-dependent P2_cas regulation.**
**A** Presence (blue: P1+) or absence (red: P1Δ) of the constitutive P1_cas promoter in the GBS population. The number of genomes analysed in each clonal groups (CG) is given in brackets. **B** Diversity of the CRISPR-Cas9 locus in the population. Left: core-gene phylogeny of representative isolates of the major sequence lineages (SL). Right: co-phylogeny of Cas9 variants. The percentage of identities (%) for P1+ (blue), P1Δ (red), and Cas9 (grey) in the population is indicated by the colour gradient. **C** Quantification of gene expression in Δ*covR* mutants by RT-qPCR relative to WT strains. Transcription of the *cas9* and *csn2* genes were quantified in the P1Δ strains BM110 (red) and COH1 (SL-17; light red), and in the P1+ strains NEM316 (SL-23; blue), 515 (SL-23; dark blue), 2603 V/R (SL-110; grey), and A909 (SL-7; light grey). Transcription levels of the CovR-regulated virulence gene *cylE* are provided. Data represents the mean +/− SD of biological replicates (*n* = 3). Significant *P*-value from multiple two-tailed unpaired t-tests (FDR 1%, two-stage method of Benjamini-Krieger and Yekutieli) against the corresponding WT strain are highlighted with stars (****$P < 0.0001$; ***<0.001; **<0.01; * <0.05; ns, not significant). Source data, including exact *P*-value, is provided as a Source Data file. **D** Quantification of *cas* transcription in Δ*covR* mutants relative to WT strains by RNA-seq. Dots represent the mean fold change of biological triplicate (*n* = 3) for the four genes of the *cas* operon in the Δ*covR* mutants of strains NEM316 (blue), 2603 V/R (grey), and A909 (light grey). Statistical analysis with DESeq2 uses the Wald test to compute *p*-values, followed by Benjamini-Hochberg correction to adjust for multiple comparisons. Raw count, normalization, and adjust *P*-value are provided in Supplementary Data S2.

promoter is the main determinant limiting CovR-dependent transcriptional activation of CRISPR-Cas9. To gain confidence, we analysed *cas* genes transcription by RNA sequencing in three WT backgrounds (Supplementary Data S2). The four *cas* genes were significantly activated by a factor of 1.5 to 2.5 in the NEM316, 2603 V/R, and A909 Δ*covR* mutants (Fig. 5D), showing the conservation of CovR repression in the GBS population.

To test the effect of CovR inactivation on immunity in a non-SL17 strain, we used the NEM316 strain, which contains an array of 14 spacers. In the WT strain, vectors containing protospacers numbered 1, 3, and 13 were efficiently eliminated (Fig. 6A). As observed previously in BM110, the vector containing protospacer 14 corresponding to the last spacer of the array was not recognized and cleaved (Fig. 6A). Absence or compromised immunity was also observed when using internal spacers at positions 5, 6, 9, and 11 (Fig. 6A). The presence of single nucleotide polymorphisms (C to T) in the interspaced repeated sequences 5 to 11 suggests that the corresponding guide RNAs are less efficiently matured compared with spacers localized between canonical repeats (Fig. 6A). In all cases, deletion of *covR* restored or improved the immune response against suboptimal internal protospacers and against the older protospacer (Fig. 6A).

Single mismatches introduced in protospacer 3 at positions 5 to 10 abolished or strongly diminished immunity ($-3 <\text{Log2 } R_T < 0$) in the WT strain (Fig. 6B). Deletion of *covR* restored or conferred strong immunity ($\text{Log2 } R_T < -8$) against mutations at positions 7 to 10, but not against mutations at positions 5 and 6 (Fig. 6B). Lastly, we tested the acquisition of new immune memory. Spacer acquisition was faster in the Δ*covR* mutant compared to the WT parental strain (Fig. 6C), although it took more subcultures to observe the effect than in the BM110 background.

Overall, the P2$_{cas}$ promoter enhances immunity upon CovR inactivation in the P1$_{cas}$-containing NEM316 strain. However, the increase in immune efficiency is attenuated compared to the BM110 strain, which aligns with the attenuated transcriptional activation of *cas* genes observed between SL-17 and non-SL-17 strains. To further investigate the relationship between the P1$_{cas}$ and P2$_{cas}$ promoters, we deleted the P1$_{cas}$ promoter in the NEM316 strain. In the ΔP1$_{cas}$ mutant, *cas* gene transcription decreases 3- to 4-fold (Fig. 6D), and immune efficiency is almost abolished (Fig. 6E). The additional deletion of *covR* derepresses the P2$_{cas}$ promoter, restoring *cas* transcription to slightly above WT levels (Fig. 6D) and concurrently restoring Cas9 immunity to slightly above WT levels (Fig. 6E). These results demonstrate that CovR provides a similar level of regulation of the P2$_{cas}$ promoter in both the NEM316 and BM110 strains. However, the presence of the constitutive P1$_{cas}$ promoter dampens *cas* gene upregulation and the increase in immune efficiency driven by P2$_{cas}$ activation when CovR is inactivated.

## Discussion

Immune systems need to be tightly regulated to ensure protection while preventing autoimmunity. Here, we describe the regulation of the prototypical type II-A CRISPR-Cas9 system by the main repressor of virulence CovR. When repurposed for gene editing, the level of Cas9 expression is one of the most critical parameters for avoiding toxicity and off-target side effects in both eukaryotic and prokaryotic cells[33–36]. This drawback is, in fact, advantageous for the natural bacterial host. By targeting a broader range of sequences, the bacterial immune response remains effective against rapidly evolving foreign elements that would otherwise escape immunity.

Transcription of *cas9* is primary dependent on the constitutive P1$_{cas}$ promoter. As described for a GAS clonal population, the Cas9-dependent increased mutation rate at the P1$_{cas}$ promoter creates a subpopulation that overexpresses Cas9, providing robust and pre-emptive defence against infections[3–6]. In contrast to a population-based strategy, the CovR-regulated P2$_{cas}$ promoter enhances single-

cell immunity. The CovRS system is the master regulator of virulence, directly repressing major virulence factors localized both in the core and in the accessory genomes[21]. Importantly, CovR functions as a transcriptional repressor, and its regulation occurs through inactivation—an inverse regulatory logic compared to most two-component systems. The P2$_{cas}$ promoter is therefore strongly repressed by CovR in standard growth condition. Activation of CovR signalling, meaning inactivation of the CovR repressor, leads to virulence gene expression and increases Cas9 immune efficiency.

CovR inactivation is a dynamic process integrating stress and cell cycle signalling, mediated by the CovS-Abx1 membrane complex and the Stk1 serine/threonine kinase[29,37]. CovS activity is required for enhanced Cas9 immunity, suggesting that stressed cells pre-emptively enhance immune surveillance. However, full CovR inactivation—likely mediated by Stk1—appears necessary for complete derepression of Cas9 immunity (Supplementary Fig. S2). The precise mechanism of CovR binding remains to be characterized to clarify the antagonistic relationship between CovS and Stk1 phosphorylation, not only at the P2$_{cas}$ promoter but also at other similarly regulated loci[21]. Interestingly, the distal positioning and non-canonical binding of CovR at P2$_{cas}$ may have been selected to ensure a tightly regulated (2- to 4-fold) Cas9 response, minimizing toxicity. This contrasts with the broader regulatory range (10- to 250-fold) observed at virulence gene promoters such as the *cyl* operon encoding hemolysin[21]. Notably, the CovR homologue in *S. pyogenes* does not appear to bind the CRISPR-Cas9 locus, as indicated by genome-wide binding datasets[38,39]. Despite the close phylogenetic relationship between GBS and GAS, their CovR regulatory networks share few common targets[40,41], supporting the idea that CovR evolves rapidly due to its central role in host-pathogen interactions and suggesting that P2$_{cas}$ is a newly evolved, de novo CovR-regulated promoter in GBS.

We showed that CovR-dependent Cas9 overexpression restores immunity against the oldest spacers in the CRISPR arrays. The position of the spacer traces the history of past infections, with the first spacer in the array being the most recent and the last being the oldest. The position of the spacer determines the efficiency of the immune response. Declining transcription along the array[42,43] and preferential maturation of the first spacer[44] allow bacteria to prioritize the immune response against the most immediate threats, *i.e.*, the latest encountered infection[43]. In contrast, CovR-dependent Cas9 overexpression could protect from recurring infections by circumventing the loss of immune memory stored in the last spacer[45].

In addition to restoring memory, Cas9 overexpression increases off-target recognition and cleavage, stochastically bypassing the dynamic equilibrium that ensure Cas9 specificity[46–49]. In vitro, mismatches in the seed sequence primarily impair DNA binding and R-loop formation, while PAM-distal mutations result in catalytically inactive complexes[50,51]. However, our experiments based on bacterial growth do not distinguish between binding and cleavage. The basal (through P1$_{cas}$) and induced (through P2$_{cas}$) levels of Cas9 expression, along with the protospacer sequence and type of mutation, differently affect the various steps required for Cas9 activity in vivo. In P1$_{cas}$-minus strains, the CovR benefit is mainly against mutations in the binding-determining seed region containing the Cas9 cleavage sites, while P1$_{cas}$ strains appear to overcome the Cas9 catalytic conformational checkpoints. Resolving in vivo binding and cleavage benefits would require characterizing a larger set of spacers across a range of Cas9 expressions.

Intriguingly, the absence of the P1$_{cas}$ promoter corresponds to GBS lineages with the most homogeneous genomes and lower rates of recombination[32,52]. This can be related to the alternative function(s) of Cas9 in bacterial physiology and evolution[53,54]. Indeed, deleting *cas9* in GBS reduces virulence and triggers transcriptomic and proteomic changes, indicating that Cas9 may regulate endogenous chromosomal genes, potentially through strain-specific CRISPR spacers[55–57].

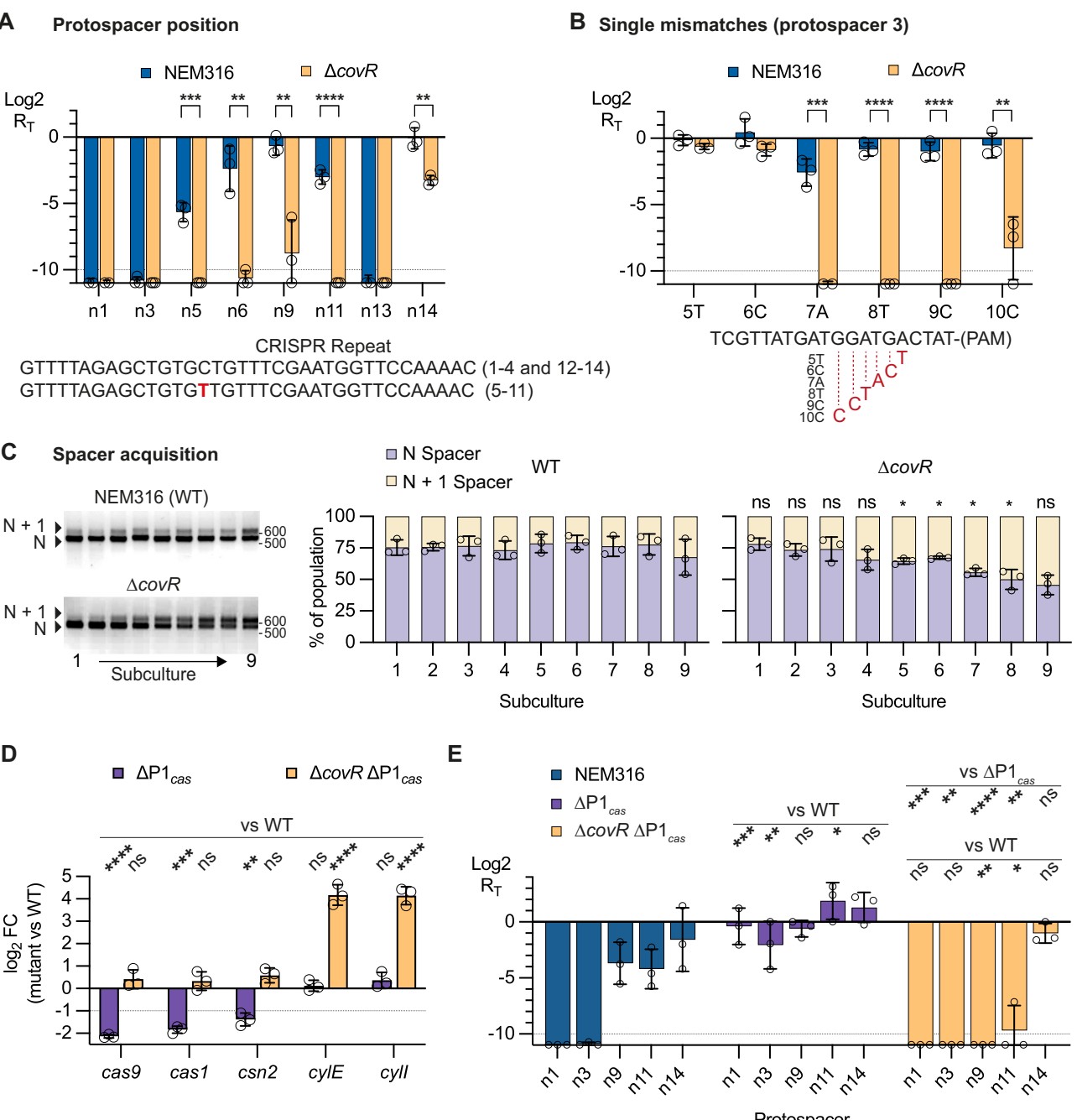

**Fig. 6 | Conservation of CovR-regulated immunity in the GBS population.**
**A** CovR inactivation increases NEM316 immunity to sub-optimal spacers. Immunity assays in NEM316 (blue) and $\Delta covR$ (light yellow) with protospacers corresponding to spacers at positions 1, 3, 5, 6, 9, 11, 13, and 14 in the NEM316 CRISPR array. The mutation (C > T) in the repeated sequence of the CRISPR array between spacers 5 to 11 is highlighted in red below the graphic. **B** CovR inactivation increases NEM316 immunity to mutated protospacers. Similar immunity assays with the n3 protospacer containing single mismatches at position 5 to 10. **C** CovR inactivation increases NEM316 immune memory. Spacer acquisition assay with representative PCRs (left) and the percentage of the population (right) having N (light violet) or N + 1 (light yellow) spacers in NEM316 and $\Delta covR$. **D** Quantification of gene expression by RT-qPCR in the NEM316 strain deleted for the P1$_{cas}$ promoter (dark

violet) and in the $\Delta covR$ $\Delta$P1$_{cas}$ double mutant (yellow). Transcription of the *cas* and *cyl* genes was quantified relative to the WT strain. **E** Immunity assays in NEM316 (blue), $\Delta$P1$_{cas}$ mutant (dark violet), and $\Delta covR$ $\Delta$P1$_{cas}$ double mutant (yellow) with protospacers corresponding to spacers at position 1, 3, 9, 11, and 14 in the CRISPR array. Deletion of P1$_{cas}$ decreases immune efficiency, while an additional deletion of *covR* restores immunity to a WT level. For all panels (**A**–**E**), bars represent the mean and error bars the standard deviation (SD) for biological triplicate ($n = 3$). Statistical significance in all panels (**A**–**E**) is determined by multiple two-tailed unpaired t-tests (FDR 1%, two-stage method of Benjamini- Krieger and Yekutieli) and highlighted with stars (****$P < 0.0001$; *** <0.001; ** <0.01; * <0.05; ns, not significant). Source data, including PCR gels and exact *P*-value, are provided as a Source Data file.

Furthermore, an analysis of spacer diversity within the GBS population revealed that over half target the species pan-genome[16,58]. In general, spacers within a given strain predominantly target prophages and conjugative elements that are present in other GBS strains or closely related streptococcal species. This has led to the suggestion that the main function of Cas9 is to modulate gene flux to maintain genetic diversity within the species, rather than being a defence system per se against rare lytic bacteriophages[16]. The extent and role of CRISPR-Cas

systems in microbial ecology and evolution, as well as the selective pressures from phages and mobile genetic elements, remain to be characterized in natural populations[59]. However, recent evidence suggests that the species-specific mobilome -comprising plasmids, integrative and conjugative elements, and prophages - are indeed the primary targets of CRISPR-Cas systems[60-63].

In conclusion, our study demonstrates the integration of CRISPR-Cas9 immunity into the regulatory network that controls host-pathogen interactions. The $P2_{cas}$ distal promoter highlights the evolution of CRISPR-Cas9 from an autonomous to a regulated defence system, capable of counteracting escape mutations in foreign DNAs and fostering both old and new immunological memories. Beyond its role in immunity, Cas9, which originated from MGEs[64], may have retained an ancient function in regulating genome and MGE dynamics. The co-regulation of Cas9 and virulence factors could have played a crucial role in shaping microbial evolution by modulating genetic variability and adaptation processes, contributing to the emergence of host-adapted virulent clones.

## Methods

### Bacterial strains
Wild-type GBS strains BM110, COH1, NEM316, 515, 2603V/R, and a rifampicin-resistant derivative of A909 are used as representative human isolates with sequenced genomes[52,65-67]. Cultures are done in Todd Hewitt supplemented with 5% yeast extract and 50 mM Hepes pH 7.4 (THY) at 37 °C under static conditions, with erythromycin 10 µg/ml, kanamycin 500 µg/ml, or chloramphenicol 5 µg/ml when required. The $\Delta covR$ mutant and the chromosomally complemented strain in BM110 were previously described[21], as well as the $\Delta covR$ mutant in NEM316[29]. Deletion of covR in the other WT backgrounds and in the BM110 $\Delta P2_{cas}$ mutant with an inactivated $P2_{cas}$ promoter[17] was similarly done with the same pG_$\Delta covR$ deletion vector[21,29]. For molecular biology, pTCV and pNZ123 vectors are cloned and maintained in Escherichia coli TOP10 (Invitrogen) and pG1 deletion vector in XL1-blue (Stratagene). Luria Bertani (LB) medium is supplemented with erythromycin 150 µg/ml (pG1), kanamycin 25 µg/ml (pTCV), or chloramphenicol 20 µg/ml (pZN123).

### CovR binding to $P2_{cas}$
Genome-wide CovR binding was determined by ChIP-sequencing, as previously reported[21], with corresponding raw data and statistical analysis available in the Gene Expression Omnibus database (NCBI, GEO database, accession number GSE158046). Electrophoretic mobility shift and DNase I protection assays were performed in vitro to assess CovR binding to $P2_{cas}$, in parallel with previously reported experiments at other CovR-regulated promoters[21]. Briefly, the purified recombinant CovR protein is incubated with [γ−32P]-dATP radiolabeled PCR probe encompassing the $P2_{cas}$ promoter (0.1 µg/µl), Poly(dI-dC) (Pharmacia), and 0.02 µg/µl BSA in binding buffer (25 mM Na2HPO4/NaH2PO4 pH 8, 50 mM NaCl, 2 mM MgCl2, 1 mM DTT, 10% glycerol) for 30 min at room temperature. Samples are separated onto a 5% TBE-polyacrylamide gel for 90 min (EMSA) or treated with DNaseI before electrophoresis (footprint) and autoradiography[21].

### RT-qPCR and transcriptional fusion
Total RNAs were purified from exponentially growing cultures ($OD_{600} = 0.5$) in THY at 37 °C, using biological triplicates for each strain. Bacteria were centrifuged at 4 °C and washed with pre-cold PBS containing RNA stabilization solution (RNAprotect, Qiagen) before flash freezing of the pellets and storage at −80 °C. Bacterial pellets were mechanically lysed with 0.1 µm beads (Precellys Evolution, Bertin Technologies) in RNApro reagent (MP Biomedicals). After beads removal by centrifugation, total RNAs were purified by chloroform extraction and ethanol precipitation. Residual genomic DNA was removed (TURBO DNase, Ambion), and RNA were quantified (Qubit

RNA HS, Invitrogen) and validated for quality (Agilent Bioanalyzer 2100). Reverse transcription (iScript cDNA Synthesis Kit, Bio-Rad) and qPCR analysis were done following manufacturer's instructions (SSO-Fast EvaGreen Supermix, Bio-Rad) with specific primers (Supplementary Data S3). Transcript levels were normalized relative to the gyrA housekeeping gene.

For promoter activity, the $P2_{cas}$ or $P_{tracr}$ promoters were amplified by PCR and cloned into the transcriptional reporter vector pTCV-lac[68] by enzymatic restriction and ligation, leading to a transcriptional fusion with the lacZ reporter gene encoding β-galactosidase. Vectors were introduced in GBS by electroporation with erythromycin selection. Promoter activities are quantified by Miller assays on total protein extracts from exponentially growing bacteria ($OD_{600} = 0.5$) in THY with erythromycin, using ONPG as substrate and $OD_{420}$ over time as readout.

### Cas9 expression
In the absence of a specific Cas9 variant antibody, we integrated a 3xFLAG encoding sequence at the 5′ end of the chromosomal cas9 gene. Briefly, three overlapping PCR were done to construct the pG_cas9::FLAG vector containing the 3xFLAG epitope flanked by 500 bp of targeted chromosomal sequences (Supplementary Data S3). Transformation in GBS, integration at the cas9 loci by homologous recombination, followed by de-recombination and loss of the vector, was done following pG-based mutagenesis procedures[21,29]. In-frame integration of the epitope was confirmed by Sanger sequencing of the targeted loci.

Bacterial cultures were grown until mid-exponential phase, centrifuged, and pellets mechanically lysed in PBS containing 1 M Dithiothreitol (DTT). Total protein content was quantified using the Bradford assay. Equal amounts (5 µg) of protein were separated by electrophoresis (4–15% Mini-PROTEAN TGX polyacrylamide gel, Bio-Rad) and transferred to a nitrocellulose membrane. Immunoblotting was done using primary monoclonal ANTI-FLAG M2 antibody produced in mouse (Sigma-Aldrich, F3165), and secondary anti-mouse IgG coupled to peroxidase produced in rabbit (Sigma-Aldrich, A9044). Chemiluminescent detection was performed using SuperSignal West Pico PLUS Chemiluminescent Substrate (Thermo Scientific), and images were acquired and analysed (PXi, GeneSys software, Syngene). Loading control was done by Coomassie staining of total proteins after electrophoresis.

### Vectors for CRISPR-Cas9 immunity
Complementary oligonucleotides (2.5 µM) with EcoRI-BamHI overhangs (Supplementary Data S3) were annealed in 10 mM Tris pH 7.5, 50 mM NaCl, 2 mM Mg acetate for 3 min at 96 °C, followed by a decrease of 0.1 °C by second until 20 °C. Diluted adaptors (20 x) were cloned in the EcoRI-BamHI digested and gel purified pTCV vector with 5000 units of T4 DNA ligase (HC, New England Biolabs) incubated at 16 °C overnight, followed by transformation in TOP10 E. coli strain with kanamycin selection. Cloning of individual protospacers was confirmed by Sanger sequencing (Eurofins) after transformant isolation and plasmid purification.

For bulk cloning, individual adaptors were pooled before cloning in the pTCV vector and used for four independent ligation reaction and E. coli transformations. Approximately $3.5 \times 10^3$ individual kanamycin resistant clones were pooled, washed, and stored in 20% glycerol at −80 °C. Cloning efficiency was first evaluated by isolating 24 randomly clones and Sanger sequencing, and validated by Illumina sequencing of all inserts in the pool of vectors (>90% of vectors with randomly distributed protospacers).

Plasmid purification from individual clones was done starting from 7.5 ml of overnight cultures following manufacturer's instructions for large (>10 kb) vector (Quiaprep Spin Minipreps, Qiagen), including doubling the standard volume of reagent, an additional

washing step in buffer PB, and DNA elution with prewarmed 70 °C water. To recover and purify the pool of plasmid, 50 µl of −80 °C glycerol stock was inoculated in 10 ml of LB with kanamycin, grown for 4 h at 37 °C, and then diluted 100 times in fresh LB with kanamycin. Plasmid purifications were done as for individual clones after overnight growth at 37 °C. Vector concentrations were quantified with fluorescent dye (Qubit, Invitrogen).

### Individual immunity assays

Transformation of GBS with pTCV vectors was done by electroporation. Competent cells were prepared starting from overnight cultures used to inoculate (1/20) prewarmed THY media at 37 °C. When reaching mid to late exponential phase ($0.5 < OD_{600} < 1$), one volume of pre-warmed hyper-osmotic buffer (1 M sucrose, 0.66 M glycine) was added and incubated for an additional hour at 37 °C. Bacteria are centrifuged (Eppendorf 5430 R, pre-cooled 4 °C, 5 min, 3000 g) and pellets washed two times (500 mM sucrose, 7 mM $NaH_2PO_4$, 1 mM $MgCl_2$, pH 7.4, pre-cooled 4 °C) in decreasing volume. Pellets were resuspended (1/50 to 1/100 volume of the initial culture) in washing buffer supplemented with 20% glycerol and immediately used or stored at −80 °C.

Aliquots of 50 µl of competent cells were mixed with 200 to 500 ng of vector (Vmax = 5 µl) and electroporated in 0.2 cm pre-cooled cuvette (2.5 kV, 25 µF, 200 ohms, Gene Pulser Xcell, Bio-Rad; or 2.5 kV Eporator, Eppendorf). After electroporation, pre-warmed THY (950 µl) was directly added in cuvettes and cells were recovered for one hour at 37 °C without agitation. Bacteria were plated (80 µl and the centrifuged remainder) on THY with kanamycin (500 µg/ml). The number of kanamycin-resistant clones was determined (Protos 3, Symbiosis) after 24 h of incubation at 37 °C. Each experiment included a control with the empty pTCV vector (without protospacer) to normalize transformation efficiency and account for differences between electrocompetent cell preparations. Immunity index is calculated as the ratio of the transformation efficiency (transformants by µg DNA) obtained with plasmid X and that of the empty control vector. At least three transformations with independent electrocompetent cell preparations have been done for each tested vector.

### Pooled immunity assays and analysis

The pool of plasmid (Input pool) contains all possible single mismatches in protospacers 4 and 8, as well as three control vectors with random protospacer sequences added to the pool (0.5 µl each for 300 µl total) before GBS transformation by electroporation. For each strain and biological replicate, five to ten transformations were done to give between 1.5 to 6.8 ×10³ kanamycin-resistant clones after 24 h of incubation at 37 °C. GBS transformants were pooled from the plates, washed with PBS, and resuspended in 20% glycerol (1.5 to 6 ml) for −80 °C storage.

Frozen aliquots (150 to 800 µl) were thawed, centrifuged, and resuspended in 120 µl of water and 600 µl of binding buffer PB (Qiagen). GBS cell wall was lysed by bead beating (0.1 µM beads, 3 × 30 sec, 8000 rpm, Precellys Evolution, Bertin Technologies) in 2 ml screwed tubes, followed by centrifugation (5 min, 4 °C). Cleared lysates were loaded on plasmid purification column (Qiaprep spin miniprep, Qiagen) and plasmid purified following manufacturer instructions with a final elution in 50 µl prewarmed 70 °C water (Output pools). Qualities were checked on agarose gels and concentration quantified by fluorescence (Qubit, Invitrogen), followed by qPCR validation of the template concentration to obtain non-saturating amplification.

Inserts containing protospacers were amplified with high-fidelity polymerase (Phusion Plus, ThermoFisher Scientific) for 25 cycles. Each vector pools (Input and Output pools and biological replicates) were amplified using pair of primers containing specific 8 bp barcodes at their 5' extremities (Supplementary Data S3). The 197 bp PCR products were purified (Qiagen), checked for quality and concentration, and pooled for multiplex sequencing. Amplicon were sent to provider (Novogen) for Illumina adaptor ligation and sequencing (PCR-free libraries, PE-250, NovaSeq).

Raw sequencing data (fastq) after demultiplexing (for Illumina adaptors by provider, then sample-specific barcodes using a custom Python script) are publicly available in the repository GEO (GSE269473). Protospacer abundance in each sample was quantified for the forward strand using 2FAST2Q v 2.6[69], with a minimal default Phred score of 30 and a perfect match approach to the protospacer sequence list containing all possible single mismatch (Supplementary Data S1 and Supplementary Data S4). On average, 158,180 reads +/− 42 410 (min 84,600 – max 216,477) were retrieved for each sample. Absolute count was used as input for a differential enrichment analysis using the R package DESeq2[70]. The enrichment or depletion of each variant is assessed in the WT compared to $\Delta covR$ in terms of a log₂FC, accompanied by a significance call using false discovery rate (FDR)-adjusted P-values, obtained through the Benjamini-Hochberg procedure. An adjusted p-value lower than 0.05 was considered significant (Supplementary Data S1 and Supplementary Data S4).

### Spacer acquisition assays

Spacer acquisition assays were performed using plasmid challenges[17] with the pNZ123 high-copy number vector. The vector is first introduced in GBS cells by electroporation, and transformants were selected and isolated on THY containing chloramphenicol. Following confirmation of pNZ123 presence by PCR, isolated transformants were inoculated in 10 ml THY without antibiotic, grown at 37 °C, and serially subcultured twice a day, for five days, without antibiotic pressure. Acquisition and quantification of spacer acquisition were done by PCR amplification of the leader end of the CRISPR array using 2 µl of each culture and primers OPL_100 and OPL_102 or OPL_105, depending on the strain (Supplementary Data S3). The relative intensity of each PCR product was measured using ImageLab software (Bio-Rad) after electrophoresis in a 2% agarose gel. Unsaturated images were converted to grayscale, and densitometry analysis was performed on each lane with local background correction to measure the intensity of the N and N + 1 fragments. The percentage of the population with N spacer was calculated as the relative proportion: intensity of band N / (intensity of band N + intensity of band N + 1).

### In silico analysis of CRISPR-Cas9 within the GBS population

A dataset of 1069 genomes representative of the GBS global population[32] was used to query for the presence of the P1$_{cas}$ promoter and for Cas9 phylogeny. Presence of either a P1$_{cas}$-positive or a P1$_{cas}$-negative genotype was tested with BLAST+ v2.15.0, with a minimum threshold of ≥90% sequence identity and ≥99% coverage, using as queries the tracRNA-cas9 intergenic region of NEM316 (389 bp region containing the constitutive promoter P1$_{cas}$) and of BM110 (280 bp). Representative sublineages and clonal groups (n = 64) were selected, together with reference isolates (including NEM316, BM110, COH1, 515, 2603 V/R, A909) to construct a core gene alignment with panaroo v1.3.4[71] from which a maximum-likelihood phylogeny was inferred with IQ-TREE2 v2.3.2[72], with 1000 bootstraps and a GTR model. Amino-acid sequences of Cas9 were extracted with BLAST+ (with a minimum threshold of ≥80% sequence identity and ≥99% query coverage), using WP_001040107 as reference. Protein sequences were subsequently aligned with Mafft v7.505[73], and a Cas9 tree was obtained with FastTree v2.1.11[74]. Co-phylogeny of core genes and of Cas9 variants was generated with phytools[75].

### RNA-sequencing

RNAs were purified as for RT-qPCR except that bacterial cultures were done on different days for biological replicates (n = 3) to account for and correct batch effects. Starting from DNase-treated RNA, rRNA depletion, library construction, and sequencing were performed

according to the manufacturer's instructions (FastSelect Bacterial, Qiagen; TruSeq Stranded mRNA and NextSeq 500, Illumina). Single-end, strand-specific 75 bp reads were cleaned (cutadapt v2.10) and mapped to the corresponding WT genomes (Bowtie v2.5.1, with default parameters). Gene counts (featureCounts, v2.0.0, parameters: -t gene -g locus_tag -s 1) were analysed using R v4.0.5 and the Bioconductor package DESeq2 v1.30.1. Normalization, dispersion, and statistical tests for differential expression were performed with independent filtering. For each comparison, raw p-values were adjusted using the Benjamini and Hochberg method, and an adjusted $p$-value lower than 0.05 was considered significant (Supplementary Data S2). Raw sequencing reads are available in the Gene Expression Omnibus database (NCBI GEO accession GSE269473).

### Reporting summary

Further information on research design is available in the Nature Portfolio Reporting Summary linked to this article.

## Data availability

The pooled immunity assays and RNA-sequencing data generated in this study have been deposited in the Gene Expression Omnibus (GEO) database under accession code GSE269473. The previously generated ChIP-sequencing data are available at GEO under accession number GSE158046. The genome metadata and phylogenetic analysis generated in this study have been deposited in the Microreact database (https://microreact.org/project/6bxfgzkmt8xkmd6frxzyuv). Other data generated in this study are provided in the Supplementary Information and the Source Data file. Source data are provided with this paper.

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

## Acknowledgements

We thank Jan-Willem Veening, David Bikard, and Sylvain Moineau for their critical reading of the manuscript and their insightful discussions. This study was supported by Agence Nationale de la Recherche (ANR-22-CE15-0024) to AF, and the National Laboratory of Excellence programme - Integrative Biology of Emerging Infectious Diseases (LabEx IBEID, ANR-10-LABX-62-IBEID).

## Author contributions

A.P., M.V.M., L.D., O.S., C.L., V.R., M.G., A.F.: Investigation; C.C., E.J., R.L.: Data curation and Formal analysis; A.P., M.V.M., P.L., A.F.: Conceptualization; A.F. Funding acquisition; P.L., A.F.: Supervision; A.P., M.V.M., C.C.: Writing - original draft; P.L., A.F.: Writing – review & editing.

## Competing interests

The authors declare no competing interests.
