## [Transparent Peer Review file · Nature Communications]

The virulence regulator CovR boosts CRISPR-Cas9 immunity in Group B *Streptococcus*

Corresponding Author: Dr Arnaud Firon

Version 0:

Reviewer comments:

Reviewer #1

(Remarks to the Author)

Pastuszka and Mazzuoli have addressed most prior concerns with extensive edits and additional experiments and controls. Briefly, they use a variety of complementary and appropriate molecular genetic techniques to convincingly show a role for the CovR in the regulation of group B streptococcus CRISPR-Cas9 immunity in a reduced system. My only remaining concern is detailing the limitations of the study. Specifically, I raised concern on only demonstrating this in rich broth and not during more relevant host-associated conditions since physiologic phosphorylation or lack thereof of CovR will dictate the relevance of the proposed mechanism. The molecular details may represent a sufficient advance, as long as appropriately detailed. A specific example of this is that I further agree with a point another reviewer made that there is insufficient support for the inclusion of "virulence" in the title. Unless activity during infection of a host is demonstrated, the authors would appropriately detail "response regulatory CovR" in place of "virulence" and narrow the related claims in the abstract and text.

Reviewer #3

(Remarks to the Author)

While there are still some minor issues that weren't able to be addressed upon resubmission. Overall, I believe they have addressed the major issues presented.

Reviewer #4

(Remarks to the Author)

The authors have done an excellent job on their revised manuscript. All my concerns have been addressed, and I support publication.

Microbiology Department

25-28, rue du Dr. Roux
75724 Paris Cedex 15
France

Paris, May 16, 2025

Arnaud FIRON, Ph.D.

Phone : +33 1 40 61 36 76

Email : arnaud.firon@pasteur.fr

To: Nature Communications

Ref: Final revisions for Nature Communications manuscript NCOMMS-25-21857-T

Thank you for the opportunity to publish our work in *Nature Communications*.

REVIEWERS' COMMENTS

Reviewer #1 (Remarks to the Author):

Pastuszka and Mazzuoli have addressed most prior concerns with extensive edits and additional experiments and controls. Briefly, they use a variety of complementary and appropriate molecular genetic techniques to convincingly show a role for the CovR in the regulation of

group B streptococcus CRISPR-Cas9 immunity in a reduced system. My only remaining concern is detailing the limitations of the study. Specifically, I raised concern on only demonstrating this in rich broth and not during more relevant host-associated conditions since physiologic phosphorylation or lack thereof of CovR will dictate the relevance of the proposed mechanism. The molecular details may represent a sufficient advance, as long as appropriately detailed. A specific example of this is that I further agree with a point another reviewer made that there is insufficient support for the inclusion of "virulence" in the title. Unless activity during infection of a host is demonstrated, the authors would appropriately detail "response regulatory CovR" in place of "virulence" and narrow the related claims in the abstract and text.

We thank the reviewer and agree with the suggestion regarding the title and have revised it accordingly, replacing "*Virulence regulates and boosts CRISPR-Cas9 immunity in Group B Streptococcus*" with "*The virulence regulator CovR boosts CRISPR-Cas9 immunity in Group B Streptococcus*".

We believe that our manuscript does not overstate the findings. We have carefully balanced the discussion, highlighting the limitations of our work, including the current gaps in knowledge regarding the signals controlling CovR activity, both *in vitro* and *in vivo*, and the major gaps on the function of CRISPR-Cas9 in its natural host, either an antiphage defence system or a regulator of genome dynamic at the species level.

Reviewer #3 (Remarks to the Author):

While there are still some minor issues that weren't able to be addressed upon resubmission. Overall, I believe they have addressed the major issues presented.

We thank the reviewer.

Reviewer #4 (Remarks to the Author):

The authors have done an excellent job on their revised manuscript. All my concerns have been addressed, and I support publication.

We thank the reviewer.